# Quorum sensing control of Type VI secretion factors restricts the proliferation of quorum-sensing mutants

Charlotte Majerczyk[1], Emily Schneider[1], E Peter Greenberg[1,2]*

[1]Department of Microbiology, University of Washington School of Medicine, Seattle, United States; [2]Guangdong Innovative and Entrepreneurial Research Team of Sociomicrobiology Basic Science and Frontier Technology, South China Agricultural University, Guangzhou, China

**Abstract** *Burkholderia thailandensis* uses acyl-homoserine lactone-mediated quorum sensing systems to regulate hundreds of genes. Here we show that cell-cell contact-dependent type VI secretion (T6S) toxin-immunity systems are among those activated by quorum sensing in *B. thailandensis*. We also demonstrate that T6S is required to constrain proliferation of quorum sensing mutants in colony cocultures of a BtaR1 quorum-sensing signal receptor mutant and its parent. However, the BtaR1 mutant is not constrained by and outcompetes its parent in broth coculture, presumably because no cell contact occurs and there is a metabolic cost associated with quorum sensing gene activation. The increased fitness of the wild type over the BtaR1 mutant during agar surface growth is dependent on an intact T6SS-1 apparatus. Thus, quorum sensing activates *B. thailandensis* T6SS-1 growth inhibition and this control serves to police and constrain quorum-sensing mutants. This work defines a novel role for T6SSs in intraspecies mutant control.

*For correspondence: epgreen@ u.washington.edu

## Introduction

Bacterial quorum sensing (QS) often controls factors that benefit cells in groups; it controls cooperative activities. A well-studied type of QS common to a number of Proteobacteria species is mediated by acyl-homoserine lactone (AHL) signals. AHLs are made by members of the LuxI family of signal synthases. The AHLs can diffuse in and out of cells. At sufficient cell densities, the AHLs accumulate to concentrations needed for expression of specific genes by binding signal receptors of the LuxR family of transcriptional regulators (for reviews of QS see (*Fuqua et al., 1996*)). In some bacterial species, QS controls just a few genes and in others it controls hundreds of genes. Genes for production of secreted or excreted products are overrepresented in AHL quorum sensing regulons. These extracellular products have at least in some cases been shown to serve as public goods. Any member of a group can benefit from the public good regardless of whether that individual has contributed its share of the good (*West et al., 2006*). There is a metabolic cost associated with QS and QS-activation of gene expression. QS control of public goods has been studied in some detail in *Pseudomonas aeruginosa*. Because QS activates production of extracellular proteases, QS signal receptor mutants have a fitness advantage over the wild type when growth is on milk protein as the sole carbon and energy source (*Diggle et al., 2007*; *Sandoz et al., 2007*). The receptor mutant fitness advantage presents a conundrum; how is QS-mediated cooperation preserved?

QS itself can be considered a cooperative trait, the QS signal is a public good, and QS is susceptible to exploitation by QS mutant social cheats. In *P. aeruginosa*, three mechanisms of QS cheater restraint have been demonstrated. The first is a direct benefit-based strategy called metabolic constraint (*Schuster et al., 2013*; *Dandekar et al., 2012*); Quorum sensing activates genes coding for

**eLife digest** Bacterial cells communicate with each other by using chemical signals. The signals allow cells living in a group to coordinate their behaviors. This cooperation can help all the cells in the group, yet scientists are puzzled about how it could evolve and persist in a population. This is because individual bacteria that essentially cheat and benefit from the cooperation of the rest of the group without contributing their fair share would have a fitness advantage.

Now, Majerczyk et al. show that a bacterium called *Burkholderia thailandensis*, which is commonly found in soil, poisons neighboring freeloaders to prevent such cheating. In the experiments, bacteria with mutations that allow them to ignore the chemical signals that trigger cooperation have an advantage over cooperative bacteria when the two types of bacteria are grown together in a liquid. However, the uncooperative mutants lose their advantage when they are grown on a surface where the cooperative bacteria touch them.

To understand why, Majerczyk et al. examined which genes were active in both types of cells in these situations. These experiments showed that chemical signals released by the cooperative bacteria cause them to produce both toxins and proteins that protect against these toxins. This allows the cooperative cells to poison cheaters that they come in contact with, while suffering no ill effects themselves. This allows the cooperative bacteria, via the signaling chemicals, to police cheats and reward only other cooperative bacteria. The next step will be to determine if other types of bacteria also use this policing strategy and to identify more pairs of genes that contribute to protecting the benefits of cooperation.

public goods (such as extracellular proteases), as well as a limited set of privately utilized goods (such as a hydrolase that allows producing cells to utilize adenosine). When cells are supplied with both casein and adenosine, QS signal receptor (LasR) mutant cheats are penalized and constrained (*Dandekar et al., 2012*). A second mechanism has been called metabolic prudence (*Xavier et al., 2011*) where the QS-controlled public good is only made when it's production is not costly. Metabolic prudence minimizes the cost of cooperation. Recently, a mechanism whereby *P. aeruginosa* cooperators punish or police cheats has been described. QS competent cooperators produce cyanide and are more resistant to cyanide than QS mutant cheats, and thus the cooperators can constrain cheats by slowing their growth with cyanide (*Wang et al., 2015*). The *P. aeruginosa* policing is a rare example of this sort of activity in bacteria, and because cyanide production and resistance has some semblance to a toxin-immunity-like mechanism we hypothesis that QS activation of toxin-immunity systems might have some generality as a policing mechanism where QS bacteria can control QS mutants.

We recently used RNAseq to identify genes in the *Burkholderia thailandensis* QS regulon (*Majerczyk et al., 2014*). *B. thailandensis* has three AHL QS systems, BtaR1-I1, BtaR2-I2 and BtaR3-I3 (*Ulrich et al., 2004*; *Chandler et al., 2009*; *Duerkop et al., 2009*). Together, the QS systems control hundreds of genes. The BtaR1-I1 regulon appears to be the largest as BtaR1 regulates >150 genes (*Majerczyk et al., 2014*). We queried the transcriptomics data and discovered that expression of several genes shown to encode Type VI secretion system (T6SS) substrates, including characterized toxins and their cognate immunity or antitoxin genes, are activated by QS. Expression of a toxin and its antitoxin results in a toxin-resistant producer cell, which can deliver toxins via the secretion apparatus via cell-cell contact to a sensitive cell thereby inhibiting growth of the sensitive cell (*Russell et al., 2014*). We show that T6S provides wild-type *B. thailandensis* with an ability to constrain BtaR1 mutants in a contact-dependent manner. This supports the hypothesis that QS activation of toxin-immunity systems can serve as a policing mechanism to control QS mutants growing among QS competent wild-type cells. Our findings also reveal yet another role for T6S in bacteria. T6S has been shown to provide a competitive interspecies or interstrain advantage, and some T6Ss have been shown to play a role in bacterial virulence (*Schwarz et al., 2010*; *2014*). Here we describe intraspecific control of QS mutants. This provides a rationale for why T6S toxin and immunity genes might be under QS control.

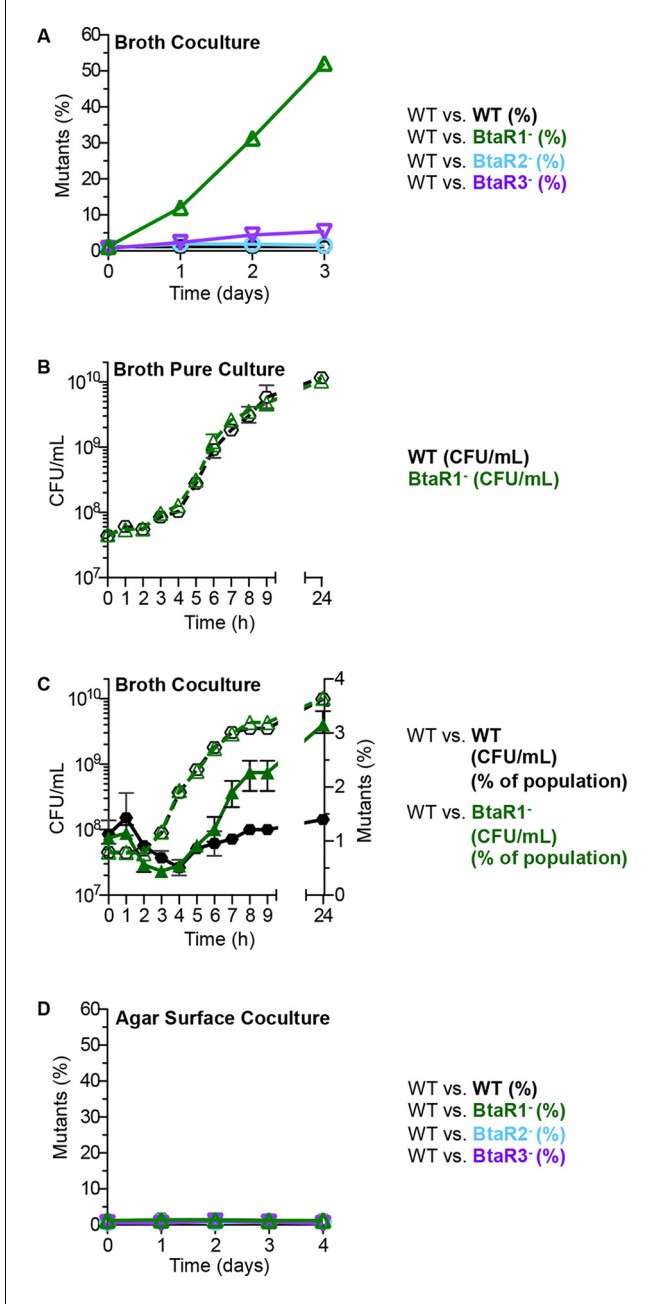

**Figure 1.** Growth of the wild type and QS mutants in LB broth and on an agar surface. (**A**) In LB broth, the wild type (strain E264) was mixed with the BtaR1- mutant CM157, the BtaR2- mutant CM159 or the BtaR3- mutant CM161 at a starting mutant relative abundance of 1%. All mutants were KmR. As a control we mixed the wild type with CM218, a KmR strain derived from the wild type. Results are percent of total colonies that are KmR (% mutants). The control shows there is little cost associated with the KmR marker. (**B**) LB broth pure culture growth curves of CM224, a TpR wild-type strain and the KmR BtaR1- mutant. Data for *Figure 1B* for the statistical analysis described in the text are in the *Figure 1—source data 1*). (**C**) 24-hour LB broth coculture. The TpR marked wild type) mixed with the KmR marked wild type), or the BtaR1 mutant. KmR bacteria were started at 1% of the population and each coculture was grown in a flask for 24 hr. Bacteria were enumerated by colony counts for total coculture yield (CFU/mL) and percent KmR mutants in the population (open symbols are CFU/mL, closed symbols are percent of mutants). Bars show the mean and range of at least three biological replicates. (**D**) Agar surface competitions. Wild type with the BtaR1- mutant, the BtaR2- mutant or the BtaR3- mutant at a starting mutant abundance of 1%. As a control we mixed the wild type with the KmR strain derived from the wild type. Data are means of at least three biological replicates. Some ranges are not visible, as they were small and within the size of the symbols.

The following source data is available for figure 1:

**Source data 1.** Experimental data for the statistical analysis for *Figure 1B*.

## Results

### The BtaR1 mutant has a growth advantage over the wild type in LB broth cocultures

To assess the fitness of the *B. thailandensis* QS mutants in competition with their parent, we cocultured the wild type with signal blind QS mutants. We used signal-blind mutants BtaR1⁻, BtaR2⁻ or BtaR3⁻ (which each have mutations in the LuxR homolog genes *btaR1, btaR2,* or *btaR3*) because they cannot respond to signals made by QS-proficient bacteria in coculture. The starting inoculum was 99% wild type and 1% signal-blind mutant. Cocultures were transferred to fresh medium at 24 hr intervals as described in the Materials and methods (there were about 12 doublings per day), and at each interval the relative abundance of parent and mutant was determined. After 3 days in broth, the BtaR1⁻ mutant showed an increase from 1% of the total population to 50% (*Figure 1A*). The BtaR2⁻ and BtaR3⁻ mutants did not have a growth advantage over the parent and remained at about 1% of the population throughout the experiment (*Figure 1A*).

Growth rates and yields of the BtaR1⁻ mutant and the parent in pure culture were similar with a small but reproducible difference during late logarithmic phase where the parent showed slightly slower growth than the mutant (*Figure 1B*). At 7 hr post-inoculation in pure liquid culture, there was a reproducible and statistically significant increase in cell number for the BtaR1⁻ mutant; the average cell yield of the wild-type parent was $1.8 \times 10^9$ CFU/mL and the BtaR1- mutant was $2.6 \times 10^9$ CFU/mL (*p* value=0.0041) (*Figure 1B*, data for this figure can be found in the *Figure 1—source data 1*). The slight difference in post-exponential growth rate coincides with the point during growth when we expect QS gene expression to be most evident, and when the BtaR1⁻ mutant begins to take over the population (*Figure 1C*). We attribute the slight influence on growth of the parent to the cost of BtaR1-mediated QS and QS-dependent gene expression. Previous experiments showed that BtaR1 controls aproximately 164 genes in post-exponential phase, whereas BtaR2 and BtaR3 controlled 20 and 12, respectively (*Majerczyk et al., 2014*). We suspect that the larger BtaR1 regulon may impart a larger burden on growth than either the smaller BtaR2 or BtaR3 regulons.

### The BtaR1 mutant does not have a growth advantage over the wild type in solid surface-grown cocultures

In contrast to broth cocultures where the BtaR1⁻ mutant had a strong fitness advantage, the BtaR1⁻ mutant had no advantage on agar plate grown cocultures at 30°C (*Figure 1D*) or 37°C (data not shown). The starting and final percent of the BtaR1⁻ mutant after four daily plate transfers (about 36 doublings in total) was 1% of the total *B. thailandensis* population. The BtaR2⁻ and BtaR3⁻ mutants had no fitness disadvantage on agar plates, as was true for broth cocultures (*Figure 1D*).

### QS regulation of known and predicted T6S effector and immunity genes

Among the many growth conditions that vary between liquid- and agar-grown cocultures is a difference in cell-to-cell contact. Cell contact is minimal during growth in broth but does occur when cells are grown on a solid agar surface. This led us to ask if QS controls contact-dependent growth inhibition systems. We previously reported that QS-1 activated the contact-dependent inhibition (CDI) system encoded by the *cdiAIB* genes (*Majerczyk et al., 2014*). However, CDI does not limit the outgrowth of the BtaR1⁻ mutant during agar-grown cocultures. We next asked whether QS regulates other contact-dependent growth inhibitory factors. *B. thailandensis* contains five different T6SSs, at least one of which is used for interspecies competition (T6SS-1) (*Schwarz et al., 2010*). The T6SS-1 comprises a pathway that delivers effector proteins to neighboring cells upon direct cell contact. Some T6SS-1 effectors have been characterized and are toxic (*Russell et al., 2014*). *B. thailandensis* is immune to the toxins because it produces cognate immunity proteins (*Russell et al., 2014*). We asked if the T6SS-1 was under QS-control by querying *B. thailandensis* QS transcriptomics data (*Majerczyk et al., 2014*) for genes for the T6SS-1 apparatus or for previously identified substrates (*Russell et al., 2012*). Although genes for the T6SS-1 apparatus are not QS-controlled, we observed that QS activates nine genes for T6SS-1 substrates, as well as an additional eight that are either adjacent to a substrate gene, fall within the same gene cluster, or include a putative T6SS-1 immunity gene (*Figure 2A*).

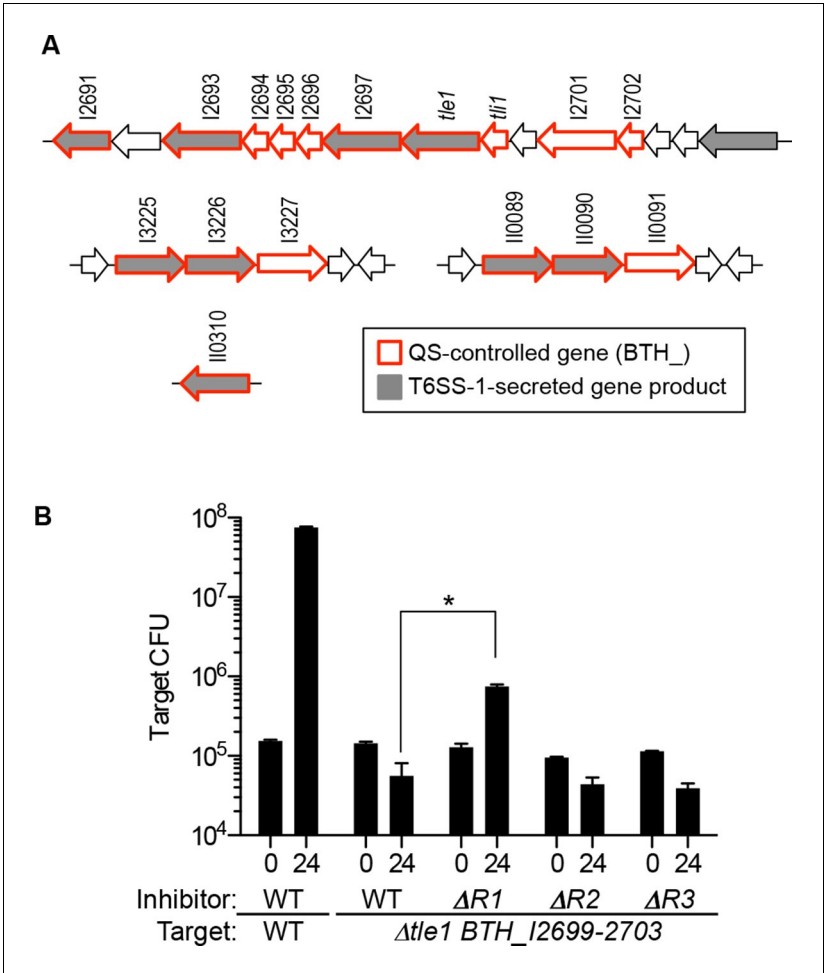

**Figure 2.** Map of QS-controlled genes for T6SS-1 substrates and solid-surface competition experiments with T6SS-1 mutants. (**A**) Genes coding for known or putative T6SS-1 substrates (*Russell et al., 2012*) are shaded grey and genes controlled by QS (*Majerczyk et al., 2014*) are outlined in red. BTH_I2698 is the *tle1* toxin gene and BTH_I2699 is the *tli1* immunity gene (*Russell et al., 2013*). The neighboring genes, BTH_I2701 and BTH_I2702 are paralogs of *tle1* and *tli1*. (**B**). Agar surface competition experiments. These experiments were started with a relative abundance of 10% target, CM316 Tp$^R$ T6SS-1 immunity mutant (△*tle1*△BTH_I2699-703) or CM218 Tp$^R$ wild type (WT). The inhibitor strains all carried a Km$^R$ marker; The wild type inhibitor was CM224, The BtaR1$^-$ mutant (△R1) was CM157, the BtaR2$^-$ mutant (△R2) was CM159, and the BtaR3$^-$ mutant (△R3) was CM161. Target cell abundance after one 24 hr round of colony growth was determined by plate counting. Data are from three biological replicates. The asterisk (*) indicates statistical significance between the CFU/mL measured in the WT or BtaR1 culture, as determined by an unpaired t test. Data can be found in the *Figure 2—source data 1*.

The following source data is available for figure 2:

**Source data 1.** Experimental data for statistical analysis of the results in *Figure 2*.

## QS mutants show a T6SS-1-mediated killing defect

Ten of 15 genes in a cluster spanning BTH_I2691-BTH_I2705 were QS activated (*Figure 2A*). Two of these 10 QS-activated genes, BTH_I2698 (*tle1*) and BTH_I2699 (*tli1*), are a known T6S toxin and immunity pair that code for a lipase and immunity protein (*Russell et al., 2013*). When a *B. thailandensis* recipient cell does not produce the Tli1 immunity protein, Tle1 acts as a T6SS-1-delivered antibacterial factor (*Russell et al., 2013*). In addition to *tle1* and *tli1*, QS activates neighboring *tle1* and *tli1* paralogs (BTH_I2701 and BTH_I2702) (*Figure 2A*). To test the hypothesis that QS activates T6SS-1-mediated killing in *B. thailandensis*, we competed the wild type or the signal blind QS mutants against a *B. thailandensis* mutant lacking *tle1*, *tli1*, and adjacent immunity paralogs

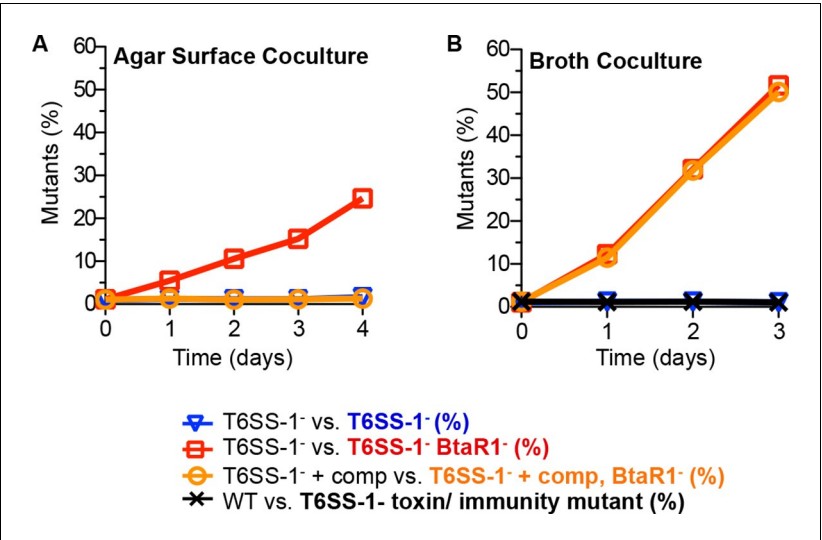

**Figure 3.** The contributions of T6SS-1 in policing BtaR1⁻ mutants. Competitions were on (**A**) LB agar plates for solid surface cocultures or in (**B**) LB broth for liquid cocultures. The T6SS-1⁻mutant (△*clpV*) was mixed with the Tp^R BtaR1⁻ T6SS-1⁻ mutant ES9 (open red squares) or the Km^R T6SS-1⁻ mutant ES18 (open blue downward triangle). The *clpV* complemented T6SS-1- mutant △*clpV*-comp was mixed with the *clpV* complemented BtaR1- T6SS-1- mutant ES16 (open orange circles). For liquid cultures, the wild type (strain E264) was mixed with CM316 Tpʳ T6SS-1 toxin/ immunity mutant (△*tle1*△BTH_I2699-703) (black crosshatch). This served as a control to demonstrate T6SS-1-mediated killing does not occur in liquid broth cocultures. Results are the mean of at least three biological replicates. Ranges were within the size of the symbols.

(△*tle1*BTH_I2699-2703). Consistent with a previous publication (***Russell et al., 2013***), the wild type outcompeted the T6SS-1 effector and immunity mutant (***Figure 2B***). We also found that the BtaR1⁻ mutant was impaired in inhibition of the effector and immunity mutant (***Figure 2B***). The BtaR2⁻ and BtaR3⁻ mutants each showed wild-type levels of inhibition (***Figure 2B***). We propose that the T6SS-1-mediated killing defect in the BtaR1⁻ mutant results from its inability to fully activate expression of one or more T6S toxin genes.

## The ability of wild-type *B. thailandensis* to compete with a BtaR1⁻mutant during agar growth depends on T6S

We hypothesized that the QS-activated T6SS-1 effector and immunity proteins in wild-type bacteria restrict the proliferation of BtaR1⁻ mutants upon cell contact in cocultures. This hypothesis was tested in LB agar competition experiments with a △*clpV* T6SS-1 secretion apparatus mutant (T6SS-1⁻) and a BtaR1⁻T6SS-1⁻ mutant. In this experiment, the BtaR1⁻T6SS-1⁻ mutant rose in frequency from about 1% to nearly 25% of the population (***Figure 3A***). When the T6SS-1⁻ mutants were complemented with *clpV* to restore the function of the T6SS-1 apparatus, a *btaR1* mutation no longer conferred a coculture growth advantage (***Figure 3A***). As predicted, in broth coculture, the BtaR1⁻ T6SS-1⁻ mutant showed a strong fitness advantage during growth with the T6SS-1⁻ mutant (***Figure 3B***). This fitness advantage was nearly identical to the fitness advantage of the BtaR1⁻ mutant during coculture in broth with the wild-type parent (***Figure 1A***) and to a T6SS-1 complemented competition (***Figure 3B***).

We reason that the parent, with an intact QS system, would express quorum-activated T6SS-1 toxin and immunity proteins, whereas the BtaR1⁻ mutant should not fully induce its T6SS-1 effectors and immunity factors. As such, the BtaR1⁻ mutant would be sensitive to cell contact-dependent growth inhibition or killing by the wild type. This can be considered a policing mechanism whereby QS-proficient cooperators hinder the success of individuals incapable of cooperation.

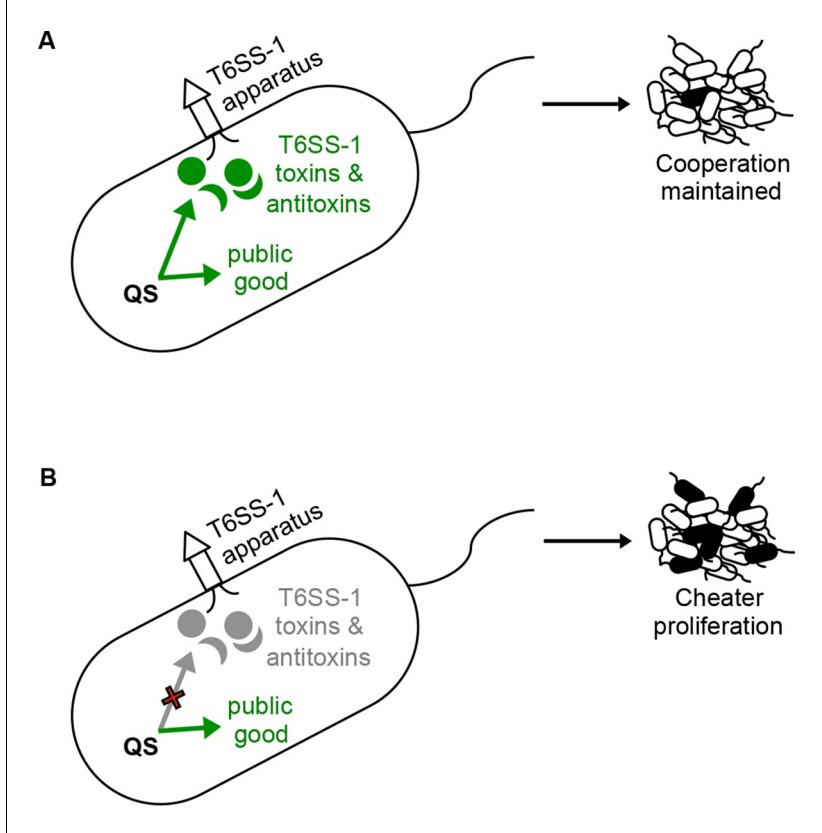

**Figure 4.** A policing model where QS control of T6S contact-dependent killing (effector and immunity proteins) enforces cooperation. (**A**) The bacterium on the left depicts a situation where QS co-regulates a public good with the T6SS-1 effector and immunity factors. Under these conditions, QS-proficient bacteria (cooperators, shown as white cells) limit the proliferation of a QS mutant (potential cheater, black cell) in a population. (**B**) In this situation, QS no longer co-regulates public good production with the T6SS-1 killing mechanism. Without co-regulation of the killing mechanism, QS mutants (potential cheaters, black cells) outcompete the wild type cooperators (white cells).

## Discussion

We present evidence that QS control of T6SS-1-mediated toxicity and immunity genes allows QS competent members of *B. thailandensis* colonies to police against BtaR1⁻ QS mutants. During plank- tonic growth in broth where T6SS-1 does not function due to limited cell contact, BtaR1⁻ mutants have a substantial fitness advantage over wild-type cells, presumably because of the cost associated with BtaR1-dependent activities. Policing of social deviants in *Myxococcus xanthus* and *Pseudomo- nas aeruginosa* has been reported previously (*Manhes and Velicer, 2011*; *Sandoz et al., 2007*). Here we describe a novel mechanism, for limiting quorum sensing mutant growth in bacterial groups.

Several functions have been ascribed to T6S. *B. thailandensis* T6S-mediated killing is active against Proteobacteria species other than *B. thailandensis* and is thought to be important for inter- species competition (*Schwarz et al., 2010*). T6Ss occur in many species of Proteobacteria where they can provide an advantage over competing species (see *Russell et al., 2014* for review). We note that in other organisms including, *Pectobacterium atrosepticum* (*Liu et al., 2008*), *P. aerugi- nosa* (*Lesic et al., 2009*), *Aeromonas hydrophila* (*Khajanchi et al., 2009),* *Yersinia pseudotuberculo- sis* (*Zhang et al., 2011*), *Chromobacterium violaceum* (*Stauff and Bassler, 2011*), *Vibrio cholerae* (*Zheng et al., 2010*), *Vibrio alginolyticus,* (*Sheng et al., 2012*), and *Vibrio parahaemolyticus* (*Wang et al., 2013*), QS regulates genes for the T6S apparatus (not the effector and immunity fac- tors as we describe for *B. thailandensis*). We suspect QS regulation of the T6S apparatus might play

**Table 1.** Bacterial strains and plasmids used in this study

| Bacterial strain or plasmid | Genotype or description[a] | Source |
|---|---|---|
| Bacterial strains | | |
| DH10B | *E. coli* cloning vehicle | Invitrogen |
| E264 | Wild-type *B. thailandensis* | (*Brett et al., 1998*) |
| JBT107 | E264 △*btaR1* | (*Chandler et al., 2009*) |
| JBT108 | E264 △*btaR2* | (*Chandler et al., 2009*) |
| JBT109 | E264 △*btaR3* | (*Chandler et al., 2009*) |
| CM157 | JBT107 *glmS1 attn7::Km*; Km$^R$ | This study |
| CM159 | JBT108 *glmS1 attn7::Km*; Km$^R$ | This study |
| CM161 | JBT109 *glmS1 attn7::Km*; Km$^R$ | This study |
| CM218 | E264 *glmS1 attn7::Km*; Km$^R$ | This study |
| CM224 | E264 *glmS1 attn7::Tp*; Tp$^R$ | This study |
| BT03399 | E264 *btaR1139::ISlacZ/PrhaBo-Tp/FRT*; Tp$^R$ | (*Gallagher et al., 2013*) |
| △*clpV* | E264 △*BTH_I2958* | (*Schwarz et al., 2010*) |
| △*clpV*-comp | △*clpV att Tn7-miniTn7T-Tp-S12-BTH_I2958* | (*Schwarz et al., 2010*) |
| ES9 | △*clpV btaR1139::ISlacZ/PrhaBo-Tp/FRT*; Tp$^R$ | This study |
| ES18 | △*clpV glmS1 attn7::Tp*; Tp$^R$ | This study |
| ES14 | △*clpV btaR1139::ISlacZ/PrhaBo scar*; Tp$^S$ | This study |
| ES15 | ES14 *glmS1 attn7::P$_{S12}$-clpV, Tp*; Tp$^R$ | This study |
| ES16 | ES15 *glmS2 attn7::Km*; Tp$^R$, Km$^R$ | This Study |
| △*tle1*△BTH_I2699-703 | E264 △*BTH_I2698-BTH_I2703* | (*Russell et al., 2013*) |
| CM316 | △*tle1* △*12699-2700 glmS1 attn7::Tp*; Tp$^R$ | This study |
| Plasmids | | |
| pTNS2 | R6K replicon TnsABC+D vector | (*Choi et al., 2005*) |
| pUC18T-mini-Tn7T-*Tp* | Cloning vector | (*Choi and Schweizer, 2006*) |
| pUC18T-mini-Tn7T-*Km*-FRT | Cloning vector | (*Choi et al., 2008*) |
| pFLPe2 | Flp recombinase-expressing vector; Zeo$^R$ | (*Choi et al., 2008*) |
| pUC18T-miniTn7T-Tp-S12-BTH_I2958 | *clpV* complementation contruct (*P$_{S12}$-clpV*) | (*Schwarz et al., 2010*) |

a role in policing QS mutants during polymicrobial competitions or in other scenarios where T6S might benefit the producing cell. Furthermore, we note that other types of toxin-antitoxin systems have been reported to be under control of other non-AHL-mediated quorum sensing systems, for example the *E. coli mazEF* encoded toxin-antitoxin genes are controlled by a linear pentapeptide quorum sensing signal (*Kolodkin-Gal et al., 2007*).

Under the conditions of our experiments the BtaR1-BtaI1 QS system carries with it a metabolic cost, which is evident in broth culture competition experiments and to a lesser extent in pure broth culture grow curves. The BtaR2-BtaI2 and BtaR3-BtaI3 QS systems do not appear to carry this large cost. We describe a mechanism by which wild-type populations can mitigate the cost of BtaR1-BtaI1 gene regulation by hindering the growth of QS mutants via regulation of cell-contact-dependent toxin and immunity proteins. This is a form of microbial policing; QS-proficient bacteria are outcompeted in populations of signal blind non-cooperators in the absence of the T6S cell-contact growth inhibition system. It is interesting that BtaR1 also promotes aggregation of cells under some conditions (*Chandler et al., 2009*). One can imagine that this system brings cells into close proximity where they can resist infiltration by other strains and species while at the same time protect themselves against social deviant BtaR1 mutant bacteria (*Figure 4*).

## Materials and methods

### Bacterial strains, plasmids, and growth conditions

Bacterial strains and plasmids are listed in *Table 1*. Bacteria were grown in Luria-Bertani (LB) broth (10 g tryptone, 5 g yeast extract, and 5 g NaCl per liter) supplemented with 50 mM morpholinepropanesulfonic acid (MOPS) buffer (pH 7.0) or on LB MOPS agar plates (LB MOPS plus 2% agar). Antibiotics were added at the following concentrations as appropriate: for *Escherichia coli*, 100 µg/mL trimethoprim (Tp), 25 µg/mL zeocin (Zeo), and 100 µg/mL ampicillin (Ap) and for *B. thailandensis*, 100 µg/mL Tp, 2 mg/mL Zeo, and 1 mg/mL Km. Except where indicated, bacteria were grown in broth at 37°C with shaking or on agar plates at 30°C.

### Mutant construction

To generate Tp-resistant mutants we transformed strains E264, △*clpV*, and △*tle*$^{BT}$ △12699–2700, with pTNS2 and pUC18T-mini-Tn7T-*Tp* as described elsewhere (*Majerczyk et al., 2014*) to make CM224 (E264 *glmS1 attn7::Tp*), ES18 (△*clpV glmS1 attn7::Km*), and CM316 (△*tle*$^{BT}$△12699-2700*glmS1 attn7::Tp*), respectively. To generate Km-resistant mutants we transformed *B. thailandensis* strains JBT107, JBT109, JBT110, E264, and CM183 with pTNS2 and pUC18T-mini-Tn7T-*km* to generate CM157 (△*btaR1 glmS1 attn7::Km*), CM159 (△*btaR2 glmS1 attn7::Km*), and CM161 (△*btaR3 glmS1 attn7::Km*). Genomic DNA (gDNA) from strain BT03399 was used to introduce a *btaR1* mutation into △*clpV* to make ES9 (△*clpV btaR1139*::ISlacZ/PrhaBo-Tp/FRT; Tp$^R$) by natural transformation as previously described (*Thongdee et al., 2008*). The trimethoprim resistance cassette was removed from strain ES9 to make strain ES14 (△*clpV btaR1139*::ISlacZ/PrhaBo scar) by using pFLPe2 as previously described (*Choi et al., 2008*). The *clpV* complementation construct pUC18T-miniTn7T-Tp-S12-BTH_I2958 (*Schwarz et al., 2010*) was introduced into ES14 by electro-transformation with pTNS2 as described elsewhere (8) to generate ES15 (ES14 *glmS1 attn7::P$_{S12}$-clpV, Tp*). ES16 (ES15 *glmS2 attn7::Km*; Tp$^R$, Km$^R$) was then made by transforming ES15 with pTNS2 and pUC18T-mini-Tn7T-*km*. We used PCR to confirm that the antibiotic resistance markers and complementation construct were inserted at the *attn7* site near *glmS1* or *glmS2* in all of the constructs..

### LB broth growth experiments

For single strain growth curves bacteria were grown in 15 mL of broth in 125-mL flasks with an overnight culture as inoculum to a starting optical density at 600 nm of 0.05. We sampled cultures every hour and determined cell numbers by plate counting. For competition experiments we used inocula from overnight cultures at a ratio of 99:1 as indicated. As above, the starting optical densities were 0.05 in a volume of 15 mL in a 125-mL flask. Abundances of each strain were determined by plate counting on selective media.

### Long-term *B. thailandensis* competition experiments

Logarithmic phase cultures of each competing strain were diluted in fresh LB medium to an optical density of 0.2. The diluted cultures were then mixed at a 99:1 or 9:1 ratio where indicated and 20 µl of the mixture was spotted on an agar plate or inoculated into 4 mL of broth in a 16 mm test tube. The solid surface-grown cocultures were incubated at 30°C and the liquid-grown cocultures were incubated with shaking at 37°C to minimize aggregation. At 24 hr intervals cells were harvested, counted and used to inoculate a new broth or plate. Colonies on plates were scraped from the agar surface, suspended in phosphate buffer (pH 7.0), and subjected to water bath sonication for 10 min to disrupt aggregates. The cells in phosphate buffer were diluted to an optical density of 0.2. Twenty µl of the density-adjusted cell suspension was spotted onto an agar plate for another round of competition. Bacteria grown in broth were similarly diluted and 20 µL of the density adjusted cell suspension was inoculated into a test tube with 4 mL broth. At each transfer cells of each of the competing strains were enumerated by selective plate counting.

## Acknowledgements

This research was supported by USPHS grants AI057141 and GM59026, and an Innovative Team Program of Guangdong Province grant 2013S034. We thank Alistair Russell and Joseph Mougous for helpful discussions.

## Additional information

### Competing interests

EPG: Reviewing editor, *eLife.* The other authors declare that no competing interests exist.

### Funding

| Funder | Grant reference number | Author |
| --- | --- | --- |
| National Institute of General Medical Sciences | GM59026 | E Peter Greenberg |
| National Institute of Allergy and Infectious Diseases | AI057141 | E Peter Greenberg |
| Guangdong Province Grant | 2013S034 | E Peter Greenberg |

The funders had no role in study design, data collection and interpretation, or the decision to submit the work for publication.

### Author contributions

CM, Conception and design, Acquisition of data, Analysis and interpretation of data, Drafting or revising the article; ES, Acquisition of data, Analysis and interpretation of data; EPG, Conception and design, Analysis and interpretation of data, Drafting or revising the article

### Author ORCIDs

E Peter Greenberg, http://orcid.org/0000-0001-9474-8041

## Additional files

### Major datasets

The following previously published dataset was used:

| Author(s) | Year | Dataset title | Dataset URL | Database, license, and accessibility information |
| --- | --- | --- | --- | --- |
| University of Washington | 2014 | Burkholderia thailandensis E264 Burkholderia thailandensis E264 Transcriptome or Gene expression | http://www.ncbi.nlm.nih.gov/bioproject/?term=PRJNA233628 | Publicly available at BioProject of NCBI (accession no: PRJNA233628) |

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
