## [Decision Letter]

Thank you for submitting your article "Quorum sensing control of Type VI secretion factors restricts the proliferation of quorum-sensing mutants" for consideration by *eLife*. Your article has been reviewed by three peer reviewers, one of whom is a Reviewing Editor. Michael Marletta has served as the Senior Editor.

The reviewers have discussed the reviews with one another and the Reviewing Editor has drafted this decision to help you prepare a revised submission.

Summary:

The main new finding of this study, which builds upon previous work from the Greenberg laboratory, is that a Type 6 secretion system (T6SS) acts as a surveillance system to eliminate social cheaters that have mutations in quorum sensing (QS), a public good. This is achieved by placing toxic T6SS-delivered effector/immunity systems under QS control.

Essential revisions:

Overall, all three reviewers thought your study was straightforward and interesting but suggest the following:

1) Although the authors show that a B.t. region coding for a number of T6SS toxin/immunity genes plays a role in this "policing" or surveillance of mutants defective for QS, it is not clear if one or more T6SS toxin/immunity genes is directly involved. This ambiguity could be addressed by ectopic expression of immunity gene(s) linked to these toxins (Tle1/Tli1 and paralogs) together with competition experiments (Figure 1). If it turns out that this experiment is not straightforward, the authors should comment on what they have tried.

2) The figures are not optimal and need the following revisions:

Figure 1 and Figure 3: Keys to each of the different strains or types of information should be provided in the Figure (especially if not all lines are defined in the figure legend as is the case for 1C).

Figure 1: What does the CFU/mL data contribute to the interpretation of this experiment? I think it would be clearer with just the mutant% .

Figure 2: This figure is so compressed that little useful information is presented. A number of T6SS effectors and co-listing of those believed to be regulated by QS BtaR1 are shown but only the BTH_12699-12703 region was examined in the study. It would be more useful to have a more complete genomic representation labeling each of the potential effectors and immunity proteins, possibly in a supplementary figure. Again the shading and colors should be defined in the figure (with more information about the parameters for considering a gene product a T6SS-1 substrate or QU-regulated provide in the legend).

Figure 2: An appropriate control would have been to use a Type VI mutant as the inhibitor versus the *tle1* mutant to demonstrate it no longer inhibits growth.

Figure 4: It is not very clear what is going on in the left panels.

Information about reproducibility need to be provided for some data panels.

---

## [Author Response]

*Essential revisions: Overall, all three reviewers thought your study was straightforward and interesting but suggest the following: 1) Although the authors show that a B.t. region coding for a number of T6SS toxin/immunity genes plays a role in this "policing" or surveillance of mutants defective for QS, it is not clear if one or more T6SS toxin/immunity genes is directly involved. This ambiguity could be addressed by ectopic expression of immunity gene(s) linked to these toxins (Tle1/Tli1 and paralogs) together with competition experiments (Figure 1). If it turns out that this experiment is not straightforward, the authors should comment on what they have tried.* We agree with the reviewer. The data demonstrate “T6S plays a role in Bt “policing” of QS-defective mutants, but it remains unknown which QS-controlled T6 effector and immunity factors are responsible for this policing activity.” Based on our current understanding of T6S and delivery of toxins to other bacteria we assume there are several secreted effectors involved in policing. This is the complication that led us away from the suggested experiments.

Our previously reported RNAseq data showed that QS controls a total of nine different proteins secreted by the T6S system, as well as an additional eight that may be associated with the activity of these secreted factors or their immunity. Of these 17 QS-controlled factors, only the activities of *tle1*and *tli1* have been characterized. For the others that are uncharacterized, we do not know which are truly toxins and which have cognate immunity factors.

The reviewers suggested an experiment to ectopically and constitutively express the *tle1* and *tli1* effector and immunity pair in the QS mutant and perform competition experiments with the wildtype. We decided against this approach because there are so many other (uncharacterized) toxin/immunity factors in the TSS-1 system. We would need to ectopically and constitutively express all QS-controlled T6S immunity factors and in doing so we would compromise the experiment by putting a heavy burden on the expressing strain. At the very least this sort of experiment would be technically challenging and require an enormous effort.

*2) The figures are not optimal and need the following revisions:*

Figure 1 and Figure 3: Keys to each of the different strains or types of information should be provided in the Figure (especially if not all lines are defined in the figure legend as is the case for 1C).

Figure 1 and Figure 3 have been modified to include keys. Please advise if further clarification is needed.

*Figure 1: What does the CFU/mL data contribute to the interpretation of this experiment? I think it would be clearer with just the mutant% .*

Figure 1 shows that the BtaR1 mutant begins to increase in a coculture with WT as the coculture population enters into post-exponential phase and stationary phase. We believe the CFU/mL data are important as they show the phase of bacterial growth. This is relevant as QS-controlled processes are regulated in a cell density-dependent manner and most are induced as cells enter post-exponential phase. We agree that it is perhaps a point for the expert, but one an expert would find problematic if missing.

*Figure 2: This figure is so compressed that little useful information is presented. A number of T6SS effectors and co-listing of those believed to be regulated by QS BtaR1 are shown but only the BTH_12699-12703 region was examined in the study. It would be more useful to have a more complete genomic representation labeling each of the potential effectors and immunity proteins, possibly in a supplementary figure. Again the shading and colors should be defined in the figure (with more information about the parameters for considering a gene product a T6SS-1 substrate or QU-regulated provide in the legend).*

In an effort to make Figure 2 more clear, we have modified the gene images and added locus tags at each QS-controlled gene. A key has also been added to supplement the information in the figure legend. We chose not to include a list of genes as this information can be found in a previous publication, where all the details of strains used and RNAseq methods and analysis are described.

Also, we highlight that many genes are QS-controlled and potentially associated with T6SS-1. However, Figure 2 examines only a subset of these, the *tle1/ tli1* toxin and immunity pair and neighboring paralogs. We hope the modified figure makes this issue more clear as well.

*Figure 2: An appropriate control would have been to use a Type VI mutant as the inhibitor versus the tle1 mutant to demonstrate it no longer inhibits growth.*

We did not include this because we already show this experiment in a prior publication (http://doi.org/10.1016/j.chom.2012.04.007).

*Figure 4: It is not very clear what is going on in the left panels.*

We have reworked both the left panel and the figure legend.

*Information about reproducibility need to be provided for some data panels.*

The legends for each figure state that data are from a minimum of three biological replicates. Please advise if any information is missing. Note that in some instances, the error bars are quite small and are not visible.